# High-Pressure Pasteurization of Soy Okara

**DOI:** 10.3390/foods12203736

**Published:** 2023-10-11

**Authors:** Amanda Helstad, Ali Marefati, Cecilia Ahlström, Marilyn Rayner, Jeanette Purhagen, Karolina Östbring

**Affiliations:** 1Department of Food Technology Engineering and Nutrition, Lund University, Naturvetarvägen 12, 223 62 Lund, Sweden; amanda.helstad@food.lth.se (A.H.);; 2Oatly AB, Ideon Science Park—Delta 5, Scheelevägen 19, 223 63 Lund, Sweden

**Keywords:** okara, high-pressure pasteurization, shelf life, food waste

## Abstract

Okara is a by-product from the production of soy beverages, which has a high content of protein and fiber. Even though it has a high nutritional value, it is generally discarded or used as animal feed or compost. The problem is its short shelf life due to its high water content and high water activity. The aim of this study was to investigate the effect of high-pressure pasteurization at 200 MPa, 400 MPa, and 600 MPa on the shelf life of soy okara. Microbiological growth, as well as thermal properties, viscosity, water holding capacity, and oil holding capacity, was evaluated after the pressure treatments. Treatment at 600 MPa significantly reduced (*p* < 0.05) the growth of total aerobic count, yeast and mold, and lactic acid bacteria for up to four weeks of storage at 4 °C. The pasting properties were increased while the water and oil holding capacities of the soy okara did not significantly change (*p* > 0.05) after high-pressure pasteurization at 400 MPa and 600 MPa. High-pressure pasteurization is therefore a potential application technique for soy okara to produce a microbiologically safe product with maintained functional properties. However, more research is needed to optimize the process and to further investigate the microbiological species present in untreated soy okara to exclude any potential food safety risks.

## 1. Introduction

The market for plant-based dairy analogs has been increasing dramatically over the last decade and is expected to grow by 10.4% until 2027, and today, the industry is currently estimated to be worth 27.3 billion USD [1]. Plant-based beverages are convenient alternatives for consumers with lactose intolerance or milk protein allergy and some consumers also choose plant-based beverages as a healthier alternative to milk due to their lower energy content and absence of cholesterol [2]. Additionally, the production of plant-based beverages has a lower environmental impact compared to dairy milk, which is an important factor for many consumers [3]. Soy beverage requires less land and freshwater use and has lower eutrophication and greenhouse gas emissions compared to dairy milk. As an example, 1 L of soy beverage generates 0.98 kg CO_2_eq compared to 3.15 kg CO_2_eq from 1 L of dairy milk [4].

In 2019, around 330 million tons of soybeans were produced in the world, where the major part was processed into animal feed, biofuel, and vegetable oil. Only 3% (9.2 million tons) were used for direct human food, such as tofu and soy beverages [5,6]. The production of soy beverages often starts with the soaking of the soybeans followed by a grinding step. The soybeans are generally cooked at around 100 °C [7] and thereafter filtered or processed in a decanter where the soy beverage is separated from a fibrous residue, also known as okara [8]. Due to the large diversity of different plant-based beverages and their subsequent okara that are available today, the okara in this paper is hereafter called soy okara for clarification. For every processed kilogram of soybeans into soy beverage or tofu, approximately 1.1 kg of soy okara is produced [9], which gives rise to approximately 10 million tons of soy okara every year on the global scale (calculated based on 9.2 million tons of soybeans processed into tofu and soy beverages annually) [5,6].

Soy okara contains about 50% dietary fiber and 25% protein [10]. It can be used as an ingredient in a wide variety of products such as bread, pasta, cookies, soups, and snacks, and can even be used as a substitute for marzipan (a confectionary ingredient based on crushed almonds or almond paste) [11]. There are also studies evaluating the effect of including soy okara in extruded foods to fortify the product with dietary fiber and protein [12,13]. Fermented food products based on soy okara such as the Japanese koji [14] and the Chinese meitauza [15] have also been explored.

Despite the potential food applications, soy okara is an industrial waste problem mainly caused by its microbial instability due to its high water content (70–80%) [16], which results in a short shelf life (about 12 h at 30 °C) [17]. The growth of *Enterobacteriaceae*, *Bacillus cereus*, yeasts, molds, and mesophilic bacteria has been confirmed [18]. However, strains of lactic acid bacteria have also been isolated and identified in soy okara, where some of them have been shown to have probiotic properties [19]. Generally, the factories producing soy beverages operate on a small scale, making it difficult to establish any larger or more profitable business with soy okara, and the upcycling of the material therefore becomes too expensive. Producers are also often dispersed over wide areas, making it difficult to collect enough soy okara for centralized processing [10]. The easiest and most cost-efficient strategy to handle soy okara has therefore predominantly been to discard it in landfills. When dumped in landfills, soy okara putrefies and generates methane, which could have been used for biogas production instead [20]. The most common solution today is the so-called first-generation recycling where the food waste is used for animal feed or composting [21]. In a life-cycle assessment of tofu, it was assumed that soy okara was used for animal feed, replacing soymeal from the feed market, and therefore contributing less than 2% of the overall greenhouse gas emissions from tofu [22].

Numerous studies describe efforts and strategies to preserve soy okara, and a common solution is dehydration to reduce water activity, which in turn inhibits microbial growth [23,24,25,26]. Thermal dehydration treatments do have a positive impact on the shelf life of soy okara [18], but the process is expensive and requires a considerable amount of energy since large quantities of water must be removed to achieve a shelf stable product. Two different patents on how to preserve soy okara without the use of drying have been issued: the first proposes the pasteurization of soy okara with the use of a scraped surface heat exchanger (up to 120 °C) [27], and the second proposes the cooling of the soy okara and its aseptic packing in the absence of oxygen [28]. However, there is an alternative pasteurization method that can be beneficial for the microbial stabilization and quality of soy okara, namely high-pressure pasteurization (HPP).

HPP is a non-thermal pasteurization technology that has the potential to inactivate microorganisms while at the same time preserving nutrients and the fresh-like quality of taste, color, odor, and texture. On an industrial scale, HPP is a relatively new unit operation for food production. It can be applied on both liquid and solid products where the material is vacuum-packed and placed in a pressure vessel where it is exposed to high pressure (400–600 MPa). The pressure is usually applied with water that is compressed with a pump and intensifier [29]. Today, there are HPP machines available with vessel capacities up to 525 L [30]. With a bulk porosity of approximately 0.2 kg/L [31], it would be possible to load at least 105 kg of soy okara in one batch, and when vacuum-packing is applied, the loading can be further increased. Due to the high viscosity of soy okara, HPP can be favorable as it does not require any pumping or stirring, which is required in other pasteurization techniques such as plate-heat exchangers or scraped surface heat exchangers. The energy requirements can also be lower compared to thermal treatments if comparing a 100 L vessel pressurized to 400 MPa (1920 kJ) with a retort containing 1 kg of saturated steam at 120 °C (2700 kJ) [29,32]. However, HPP requires a high level of investment and maintenance costs [33], which can make it financially difficult to implement this type of processing in the near-future for small-scale producers of tofu and soy beverages. However, depending on the product, packaging, and processing, HPP can be more efficient from an environmental point of view [33].

The mechanism of microbial inactivation by HPP is suggested to be damage or alterations to the cell membrane, nucleoids, ribosomes, and enzymes. The primary effect of inactivation is suggested to be damage to the cell membranes, including the membrane-bound enzymes, although the inactivation mechanisms are not yet fully understood. Microbial inactivation through HPP depends on the type of microorganisms (bacteria, yeasts, and molds), form (vegetative cells, spores, Gram-positive, or Gram-negative), strain, species, genus, and growth phase. Vegetative cells are more sensitive than spores, and Gram-negative bacteria are more sensitive than Gram-positive bacteria [29].

HPP is also a technological tool for modifying various foods as it can alter food proteins and change properties such as solubility, emulsification, gelation, water holding capacity, and digestibility [29]. Fibers can be affected by HPP treatments and there are studies on soy okara where the aim has been to increase the percentage of soluble fibers using high hydrostatic pressure (HHP) [34] or to improve the extraction yield of proteins and fibers using high-pressure homogenization (HPH) [35], which have been successful. To the best of the authors’ knowledge, there is a lack of research evaluating the effect of HPP on soy okara with the aim of controlling microbial growth.

The aim of this study was therefore to investigate the possibility of reducing the microbiological growth in soy okara using HPP to prolong the shelf life. Functional properties were also evaluated to explore how the thermal properties, viscosity, and water and oil holding capacities were affected after the HPP treatment at different pressures.

## 2. Materials and Methods

### 2.1. Materials and Chemicals

Soy okara was kindly provided by The Green Dairy (Karlshamn, Sweden) and was directly analyzed or stored in a freezer (−18 °C) until further analysis.

The microbiological analysis was performed on four different agars: Malt Extract Agar (MA) (Sigma-Aldrich, St. Louis, MO, USA), tryptic soy agar (TSA) (Sigma-Aldrich, St. Louis, MO, USA), De Man Rogosa and Sharpe agar (MRS) (Merck, Darmstadt, Germany), and Violet Red Bile Dextrose agar (VRBD) (Merck, Darmstadt, Germany). The samples were diluted in peptone water at 0.1% wt (Oxoid, Hampshire, UK).

Rapeseed oil (ICA Sweden AB, Solna, Sweden) was purchased at a local supermarket and was used for the oil holding capacity analysis.

### 2.2. Experimental Design of HPP-Process and Storage Study

The Green Dairy does not produce soy beverages daily and, due to logistical challenges, the okara needed to be frozen before the HPP treatments. Crude soy okara samples were therefore frozen (−18 °C) for at least 24 h and thereafter thawed in a water bath (starting temperature approximately 50 °C) before being vacuum-packed (Multivac A300/11, Sepp Haggenmüller SE & Co. KG, Wolfertschwenden, Germany) in 200 g batches in PA/PE bags. The packed material was transported in an insulated cooler bag with ice packs (approximately 4 h in total) to HPP Nordic (Landskrona, Sweden), the facility where the HPP was performed (High-Pressure Processor AV-10, AVURE Technologies, Erlanger, KY, USA). The samples were pasteurized for 3 min (a typical commercial holding time) [36] at three different pressures, 200 MPa, 400 MPa, and 600 MPa. The protocols with the temperature and pressure intervals are presented in Table 1. The experiment was performed with three bags of soy okara at each pressure.

Frozen and thawed untreated soy okara was stored for 4 h, 2 weeks, and 4 weeks at 4 °C, and is hereafter referred to as reference. The HPP-treated samples were stored for 2 weeks and 4 weeks under the same conditions.

To evaluate the effect of freezing on microbial growth, crude soy okara fresh from production was also analyzed. The crude sample, collected in the factory, was kept cooled (4 °C) and arrived at the lab after 4 h. Figure 1 summarizes the study design.

### 2.3. Microbiological Growth

From each sample, 10 g was transferred into a sterile bag and 90 mL of 0.1% peptone water was added. Samples were homogenized with a stomacher (Seward BA6021 stomacher, Bury St. Edmunds, UK) for 1 min, followed by a conventional dilution series with 0.1% peptone water. An amount of 100 µL was transferred to each Petri dish and was spread over the surface with sterile glass beads. Samples were plated in duplicates at each dilution level. The plates were incubated as follows: TSA was incubated aerobically at 30 °C for 72 ± 6 h (total aerobic count), MA was incubated aerobically at 25 °C for 5–7 days (yeast and mold), MRS was incubated anaerobically at 37 °C for 72 ± 6 h (lactic acid bacteria), and VRBD was incubated aerobically at 37 °C for 24 ± 1 h (Enterobacteriaceae). The acceptable numbers of colonies on each agar plate were considered to be 20–300 colonies and the lower detection limit of the analysis was 2.30 log cfu/g.

### 2.4. Proximate Composition

The soy okara reference was sent for proximate analysis to the accredited lab Eurofins Food & Feed Testing Sweden, Lidköping. The methods used were Kjeldahl (Nx6.25) for protein, NMKL 160 mod. for fat, (EU) nr 1169/2011 (calculated) for carbohydrates, AOAC 991.43 mod. for fiber, NMKL 173 for ash, and NMKL 23 for water. To investigate any changes in the ratio between soluble and insoluble fiber due to the HPP treatment, both reference and HPP-treated samples were sent for analysis on soluble and insoluble fiber (AOAC 991.43 mod.). The samples sent for analysis were measured in one replicate. The standard deviations were based on measurement uncertainty.

The dry matter (DM) content was analyzed on-site for the reference and the HPP-treated soy okara according to the AACC 44–15A method where a sample of 3–5 g was weighed in a metal container before and after drying in an oven for 16 h at 103 °C. The analysis was performed in triplicate.

### 2.5. Water Activity

The water activity (a_w_) was measured on the reference and the HPP-treated soy okara using a water activity meter (AquaLab Ver 3TE, Decagon Devises, Pullman, WA, USA) at 20 °C. The instrument was calibrated with standard salt solutions of 13.41 M LiCl (0.250 a_w_), 8.57 M LiCl (0.500 a_w_), and 6 M NaCl (0.760 a_w_). The analysis was performed in triplicate.

### 2.6. Differential Scanning Calorimetry

The thermal properties of the reference and HPP-treated soy okara were investigated through differential scanning calorimetry (DSC) (Seiko 6200 DSC, Seiko Instruments Inc. Shizuoka, Japan). Soy okara samples of 9–12 mg were weighed into aluminum pans. No additional water was added, as the water content was already around 76%, with a sample water ratio of 1:3. The pans were sealed and run in the DSC with the settings presented in Table 2. An empty pan was used as a reference. To achieve a stable baseline for all samples, a holding time of 1 min was applied. After the measurement, DM content was analyzed by puncturing and drying the pans in an oven at 105 °C until a constant weight was achieved. Analysis was performed in at least triplicate.

### 2.7. Viscosity

The viscosity of the reference and HPP-treated soy okara was analyzed using a high-temperature Rapid Visco Analyzer (RVA 4800, Perkin Elmer, Waltham, MA, USA). The test profile is presented in Table 3, and the samples were run at a moisture content of 85% (15 g sample and 10 g deionized water) in triplicate.

### 2.8. Water and Oil Holding Capacities

Water and oil holding capacities (WHC and OHC, respectively) were analyzed for the reference and the HPP-treated soy okara samples according to the method described by Aziah et al. [38] with a few modifications. Soy okara was weighed into Falcon tubes (50 mL) together with deionized water or rapeseed oil. For WHC, 10 g of soy okara was mixed with approximately 40 mL of deionized water, and for OHC, 2 g of soy okara was mixed with 20 mL of rapeseed oil. The mixtures were vortexed for 2 min, followed by incubation at room temperature for 30 min. Thereafter, the samples were centrifuged at 4900× *g* for 20 min (Beckman Coulter, Avanti^®^J-15R Centrifuge, Brea, CA, USA). The supernatant was decanted, and the Falcon tube and pellet were reweighed. The analysis was performed in triplicates. The WHC (Equation (1)) and OHC (Equation (2)) were calculated according to the following equations, respectively.
(1)$$WHC=\frac{m\left(water\;content\;in\;sample\right)+m\left(test\;tube+pellet\right)-m\left({test\;tube+sample}_{dry\;solid}\right)}{m({sample}_{dry\;solid})}\frac{ml}{g_{dry\;solid}}$$
(2)$$OHC=\frac{m\left(test\;tube+pellet\right)-m(test\;tube+sample)}{m(sample)}\frac{ml}{g}$$

### 2.9. Scanning Electron Microscopy

The reference and HPP-treated oat okara samples were freeze-dried (Labconco, MO, USA) and milled, and thereafter gently glued onto scanning electron microscopy (SEM) stubs and sputter-coated with gold (Cesington 108 auto, 120 s, 20 mA). A scanning electron microscope (SEM; Hitachi SU3500, Tokyo, Japan) at 5 kV was used to view the samples.

### 2.10. Statistical Analysis

The software Minitab was used for the statistical analysis in this study. For each data set, normality was checked. To evaluate significant differences for normally distributed data sets, a Tukey test [39] was performed, and for non-parametric data sets, a Games–Howell test [40] was performed. *p*-values < 0.05 were considered significantly different.

## 3. Results and Discussion

### 3.1. Microbiological Content

The shelf life was evaluated using the microbiological content in the reference and the HPP-treated soy okara after 2 weeks and 4 weeks at 4 °C. Enterobacteriaceae content was below the detection limit in the crude sample and was therefore not further investigated in this study. For practical reasons, the soy okara needed to be frozen before the HPP trials. The frozen and thawed reference had a significantly lower (*p* < 0.05) growth of total aerobic count and yeast and mold compared to crude soy okara (Figure 2A,B). The freezing treatment for the reference and HPP-treated samples might therefore have given an advantage in the microbiological storage study. The growth of lactic acid bacteria was low in both the reference and the crude soy okara (Figure 2C) and no significant difference could be detected (*p* > 0.05).

For all three agars, TSA (total aerobic count), MA (yeast and mold), and MRS (lactic acid bacteria), the HPP treatment of 600 MPa had a significantly lower microbiological growth (*p* < 0.05) compared to the reference and treatments at 200 MPa and 400 MPa after 2 weeks and 4 weeks (Figure 2A–C). The reference and the treatments at 200 MPa and 400 MPa almost reached a stationary phase after 4 weeks (log 7–8 cfu/g) as nutrients became limited, while the treatment at 600 MPa remained in its exponential phase (log 6–7 cfu/g). The treatment at 400 MPa had a significantly lower microbial growth of yeast and mold and lactic acid bacteria after 2 weeks (*p* < 0.05) compared to the reference and 200 MPa, but it reached the same microbial load after 4 weeks. In a study by Voss et al., the microbiological content of soy okara was investigated after thermal treatment at 80 °C and 200 °C [18]. After 15 days of storage (20 °C), the microbial load of yeast and mold increased from 2 to 8 log cfu/g for soy okara heat-treated at 80 °C for 5 h, and from 2 to 7 log cfu/g for soy okara heat-treated at 200 °C for 1 h. In the present study, the microbial load of yeast and mold increased from 2.3 to 4.6 log cfu/g for the 600 MPa treatment after 2 weeks. Therefore, HPP might have an advantage over thermal treatment in terms of inactivating yeast and mold in soy okara. However, a direct comparison is difficult since the soy okara in the study by Voss et al. was incubated at 20 °C [18], whereas the present study used 4 °C as the incubation temperature.

The exact mechanism behind HPP inactivation of microorganisms is not yet fully understood. Hypotheses in the literature state that the primary effect of microbial inactivation is due to damage to the cell membranes and restructuring of proteins. The restructuring and unfolding of proteins also affect the membrane-bound enzymes, nucleoids, and ribosomes, all critical for microbiological growth. If these functions are inhibited, the growth will be impaired [29].

It is proposed that at 100–200 MPa, the quaternary structure is affected, and oligomeric protein structures are split into monomer structures. At 300–500 MPa, an intermediate or melted state occurs and some aggregation takes place, and at pressures above 700 MPa, the proteins unfold [29].

It has been reported that only the quaternary structure of proteins is disrupted at 200 MPa due to the fact that this structure is mainly held by hydrophobic interactions that are sensitive to pressure, and microbes can therefore quickly recover [29]. This is in line with the present study where microbial growth after the 200 MPa treatment was not significantly different (*p* < 0.05) from the reference on any agar types and this HPP treatment therefore had no effect on the shelf life.

The literature about HPP and soy okara is limited, but in a previous study, high hydrostatic pressure was applied to oat and rice grains soaked in water (1:1) for 4 h, which could be considered a similar product to soy okara. With a treatment of 500 MPa for 5 min, they exhibited a decrease in total bacterial count by more than 5.8 orders of magnitude, and when treated at 600 MPa for 5 min, no microbial growth was observed [41]. This is a considerably higher reduction compared to this study, where the total aerobic count only had an approximate 2 log reduction at 600 MPa for 3 min compared to the reference. It is known that the effect of HPP is a combination of pressure, time, and temperature [29]. Therefore, both time and temperature of the high-pressure treatment could therefore be considered in future studies.

A safe food product should have limited growth of total aerobic count, with a maximum microbial load of log 5–6 cfu/g [42,43]. Soy okara, treated at 600 MPa, could therefore be considered microbiologically safe after 2 weeks of storage (log 4.6 cfu/g). However, food safety should be considered carefully as the spore-forming bacteria *Bacillus* has previously been detected in soy okara [18], which could represent a substantial part of the growth of total aerobic count in this study (Figure 2A). The growth of *Bacillus* must be limited due to its ability to produce enterotoxins, which can already cause food poisoning at low doses [18]. Bacterial spores are difficult to inactivate with HPP, but the treatment can trigger spore germination [44]. Both thermal treatments and HPP treatments alone can have difficulties in inactivating spores. However, with the right combination of heat and pressure, spore germination and, in turn, inactivation for a certain bacterial spore could be optimized [45]. In an earlier study, it was observed that the inactivation rate of *B. stearothermophilus* spores in mashed broccoli increased when combining thermal treatment (>60 °C) with HPP [45].

Viable but nonculturable (VBNC) bacteria can also be a potential food safety problem [46], which would also require further investigation to ensure a safe soy okara product.

### 3.2. Proximate Composition and Water Activity

The proximate composition of the soy okara in this study deviates from the data in the literature. From various studies, the general content has been reported to be 25% protein, 10% fat, and 50% dietary fiber [10,20,47,48], while the soy okara in this study had a higher protein (41.8 ± 4.2%) and fat (22.9 ± 2.3%) content and lower fiber content (26.9 ± 4.0%). The variations are probably due to differences in production methods and soybean cultivars [10]. The ash content was 4.2 ± 0.4% and the carbohydrate content was 4.2% (calculated by difference).

Water content and water activity are important factors for microbial growth. The water content was around 77% and the water activity was around 1 for all treatments, including the reference (Table 4) with no significant differences (*p* > 0.05).

The total dietary fiber content consisted of only insoluble fiber in the reference soy okara (Table 5). This is a different result compared to other studies where the total dietary fiber (54–55%) has consisted of 4.2–4.7% soluble fiber [20,47]. There was no increase in soluble fiber content for the 200 MPa and 400 MPa treatments; however, for the 600 MPa treatment, an increase of 1.1% soluble fiber was observed (Table 5). A redistribution from insoluble to soluble fiber has previously been observed in soy okara after high hydrostatic pressure treatment at 200 MPa and 400 MPa (at 30 °C and 60 °C), which in turn affected the water and oil holding capacities [34]. Possible reasons for the differences between the mentioned study and this study could be the different production processes of the soy okara, and the temperature difference in the HPP treatments.

### 3.3. Thermal Properties

The thermal properties in a sample can identify native proteins, starches, fibers, and other complexes. In this study, it was investigated whether the HPP treatments had any effect on the thermal properties using DSC. The water content of the samples presented in Figure 3 was between 76 and 79%.

The soy okara had generally small peaks (<1.0 mJ/mg), which were difficult to interpret. Denaturation temperatures for the main globular proteins in soy, β-conglycinin and glycinin, have previously been determined to be 68 °C and 88 °C, respectively, at neutral pH [49]. Peaks at these temperatures are not distinct in the non-treated or HPP-treated soy okara (Figure 3), which indicates that most of these proteins have already been denatured in the production process of soy beverages, where temperatures up to 100 °C can be reached [7]. The smaller and randomly dispersed peaks could represent other intermediate states of proteins or fibers.

Denaturation temperatures of soluble and insoluble fiber in soy okara are around 86.6 °C [50] and 130–150 °C [51], respectively. The peak at 132.4 °C for the reference was not present for the HPP-treated soy okara (Figure 3). A redistribution from insoluble to soluble fiber was observed for the 600 MPa treatment (Table 5), which could explain the disappearance of the peak as the fiber structure was affected. The fiber redistribution was not observed in the 200 MPa and 400 MPa treatments, but it does not exclude the possibility that the insoluble fiber might have been affected, which in turn can have altered melting temperatures.

### 3.4. Viscosity

The viscosity was examined during heating up to 140 °C followed by cooling on the reference and HPP-treated soy okara to evaluate any effects on the gelation and pasting capacities of soy okara, which can be useful information in product development.

The viscosity for all treatments started at high levels (3000–3500 cP), which was due to the physical resistance of soy okara, whereafter continuous heating reduced its viscosity (Figure 4). The reference had two small peaks in viscosity around 100 °C (2500 cP) and 140 °C (1700 cP). The soy okara treated at 200 MPa had a small viscosity peak of around 130 °C (1900 cP), while the soy okara treated at 400 and 600 MPa had larger-viscosity peaks of around 140 °C (2100 cP and 1900 cP, respectively) (Figure 4, peaks marked with arrows). As a comparison, starchy flours such as wheat, rye, and white rice have low initial viscosities (<100 cP) and generate peaks around 95 °C with viscosities ranging from 1000 to 4000 cP [52].

The soluble fiber content increased slightly for the soy okara treated at 600 MPa (Table 5), which could explain the larger-viscosity peak that appeared for that sample as soluble fiber generally has a higher viscosity and a better ability to form gels [53]. However, the 400 MPa treatment had an even larger viscosity peak but without any increase in soluble fiber. This means that the HPP might have had another unknown effect on the 400 MPa and 600 MPa treated samples that could have created this viscosity change.

Viscosity changes were observed at 140 °C, which indicates that the components are rearranged into new structures. This might be favorable in processes where viscosity changes are desired, such as in high-temperature extrusion applications.

### 3.5. Water and Oil Holding Capacity

The fiber in soy okara (reference) was altogether insoluble, as well as in the soy okara treated at 200 MPa and 400 MPa, but with a slight increase in soluble fiber in the 600 MPa treatment (Table 5). Insoluble fiber has a porous matrix that can absorb, swell, and entrap water and therefore generally has a high WHC [53]. The WHC of the reference was 6.03 ± 0.14 mL/g and it was not affected by the HPP treatments (Figure 5). The WHC was comparable to wheat bran (6.6 mL/g) and oat bran (5.5 mL/g) and was considered high. A material with high WHC can be utilized as a functional ingredient to reduce the energy content and adjust the texture and viscosity of formulated foods [54].

The reference soy okara had an OHC of 0.75 ± 0.05 mL/g (Figure 5) and after treatment at 200 MPa, the OHC was significantly reduced (0.66 mL/g) (*p* < 0.05). However, the OHC for soy okara treated at 400 MPa (0.71 ± 0.03 mL/g) and 600 MPa (0.68 ± 0.01 mL/g) did not significantly differ (*p* > 0.05) from either the reference or the soy okara treated at 200 MPa. Therefore, the HPP treatments did have a limited impact on the OHC, and the difference would probably not affect a final product based on HPP-treated soy okara. A high OHC can make an ingredient suitable for the stabilization of high-fat foods and emulsions. Compared to dietary fiber concentrates from orange bagasse (peel, albedo, and seed), 0.86–1.28 mL/g [54], the soy okara had a slightly lower OHC, but probably high enough to be able to contribute to the functional properties of a product.

### 3.6. Scanning Electron Micrographs

Micrographs of the reference and the HPP-treated samples were taken to investigate whether the pressure treatments had any effect on the structure of soy okara.

In the reference micrographs (Figure 6a,b), inhomogeneous, flakey surfaces were observed, representing the fiber structure. The treatment of 400 MPa (Figure 6e,f) and 600 MPa (Figure 6g,h) had a similar structure compared to the reference, but with more compact and rough cell structures. The micrographs of soy okara after treatment at 200 MPa (Figure 6c,d) had a different appearance with smaller structures that coated the surfaces.

## 4. Conclusions

The HPP treatment at 600 MPa significantly reduced the growth of total aerobic count, yeast and mold, and lactic acid bacteria in soy okara after four weeks of storage at 4 °C in the absence of oxygen. The soy beverage production prior to HPP treatment involved elevated temperatures, up to 100 °C, which induced protein denaturation. Therefore, the HPP process had a minor effect on protein denaturation. We suggest that HPP treatment under high pressures (600 MPa) had a larger impact on the fibers in soy okara, where insoluble fibers were partly converted into soluble fiber, which affected the pasting properties. The HPP-treated soy okara also maintained a high WHC and OHC independent of the pressure applied, which increases the utility of soy okara since such an ingredient can contribute to the texture and viscosity of a final product.

The reduction in microbial growth and increased viscosity properties through the HPP treatment at 600 MPa create opportunities for a new preservation technique of soy okara, which could eliminate the need for drying or freezing of this by-product. More studies are needed to increase the knowledge regarding which types of microorganisms are present in the soy okara. When this is known, it is possible to design a process to reduce the growth even further and to evaluate the microbiological safety of the product. If spores are present, a combination of pressure and heat treatment could be appropriate for future investigations.

HPP has significant potential in the food industry where it can assure safety and quality attributes similar to thermal treatment, while at the same time being able to keep freshness, flavor, and nutritional content intact. However, future work within this technology include increasing understanding of the effect of HPP more deeply on different operating conditions to clarify the effects toward toxicity, allergenicity, loss of digestibility, and eating quality of foods [55]. Due to the food safety problem of bacterial spores and VBNC, there is also future development to combine HPP with other thermal or other alternative treatments such as low pH or natural antimicrobials [46,55]. Studies on HPP that contribute with additional data can therefore be valuable in the development of HPP treatment within the food industry. This study contributed new information on microbial inactivation in soy okara with HPP, which has not been examined before. Due to the complex matrix of soy okara and the probable spore problem, it can be concluded that soy okara and similar products might need other processing conditions, such as longer holding time, higher pressures, or higher temperatures.

## Figures and Tables

**Figure 1 foods-12-03736-f001:**
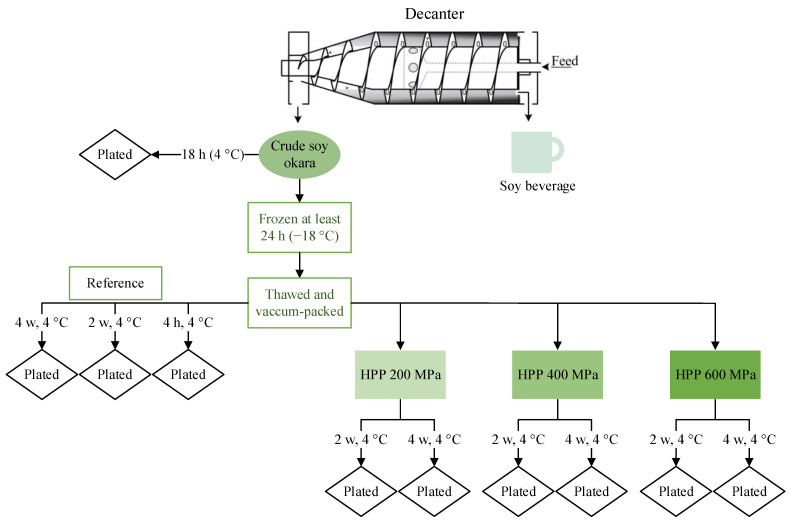
The experimental design of the microbiological storage study of HPP-treated soy okara, reference sample, and crude sample (decanter image adapted with permission from Ref. [37] 2023, Alveteg).

**Figure 2 foods-12-03736-f002:**
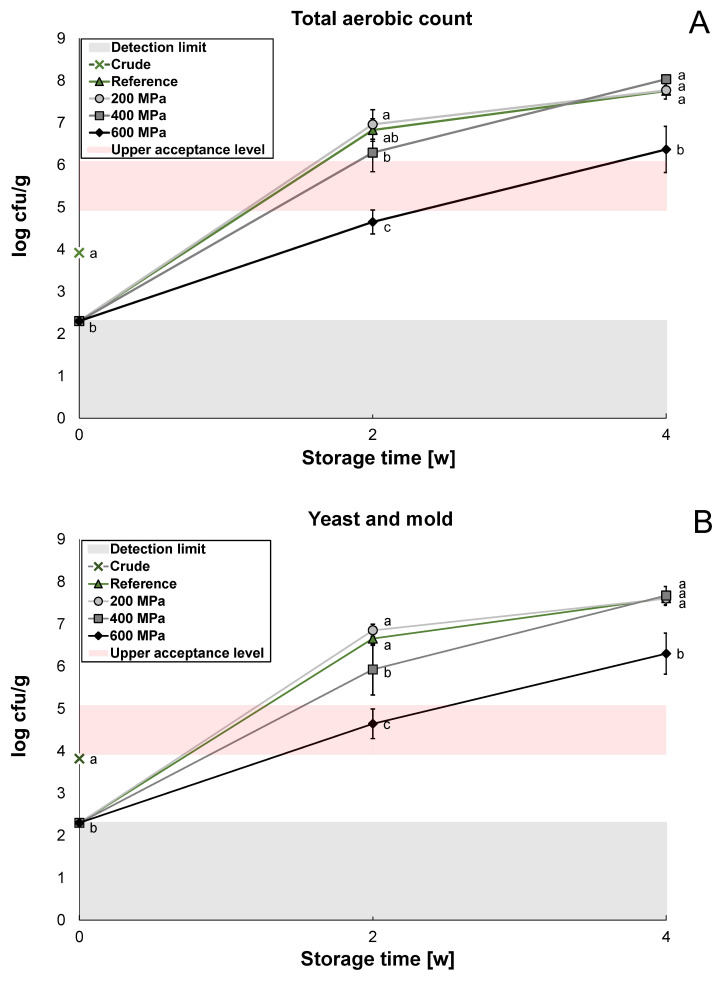
The microbiological growth of crude soy okara (cross), reference (triangle), and HPP-treated soy okara (200 MPa (circle), 400 MPa (square), and 600 MPa (diamond)) stored for 4 weeks (4 °C, vacuum-packed) on TSA (**A**), MA (**B**), and MRS (**C**). Statistical comparisons were made for crude soy okara, reference soy okara, 200 MPa, 400 MPa, and 600 MPa at each storage time for each agar type. Data with different letters are significantly different, *p* < 0.05, *n* = 6.

**Figure 3 foods-12-03736-f003:**
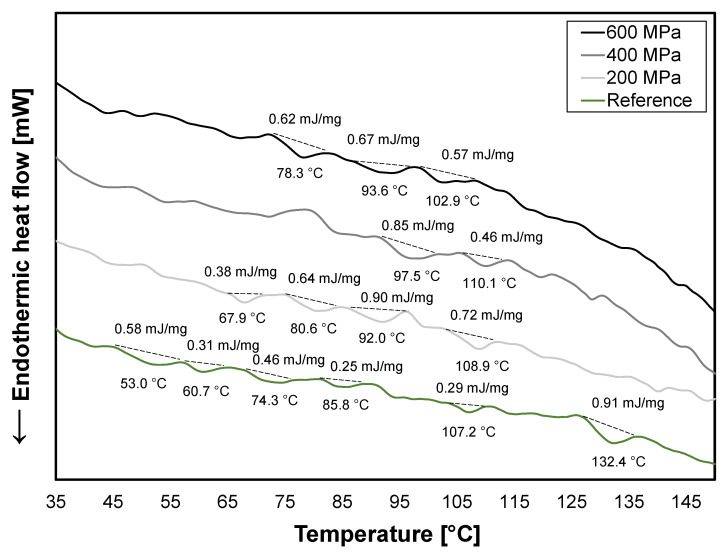
DSC curves for reference (green) and HPP-treated soy okara at 200 MPa (light gray), 400 MPa (gray), and 600 MPa (black).

**Figure 4 foods-12-03736-f004:**
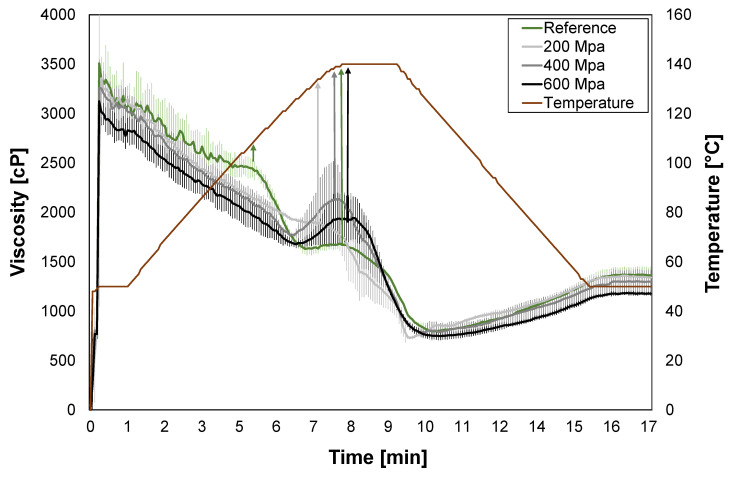
RVA graphs presenting average viscosity for reference (green) and HPP-treated soy okara: 200 MPa (light gray), 400 MPa (gray), and 600 MPa (black). Peaks are marked with arrows.

**Figure 5 foods-12-03736-f005:**
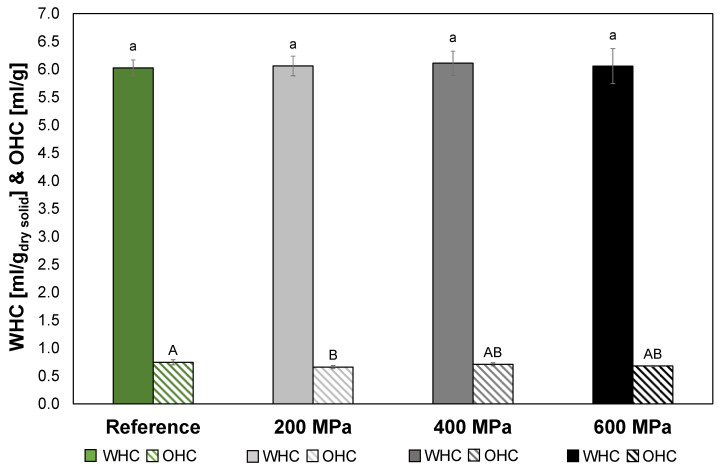
WHC (solid) and OHC (striped) for reference (green) and HPP-treated soy okara: 200 MPa (light gray), 400 MPa (gray), and 600 MPa (black). Data with different letters are significantly different, *p* < 0.05, *n* = 3 (small letters for WHC, and capital letters for OHC).

**Figure 6 foods-12-03736-f006:**
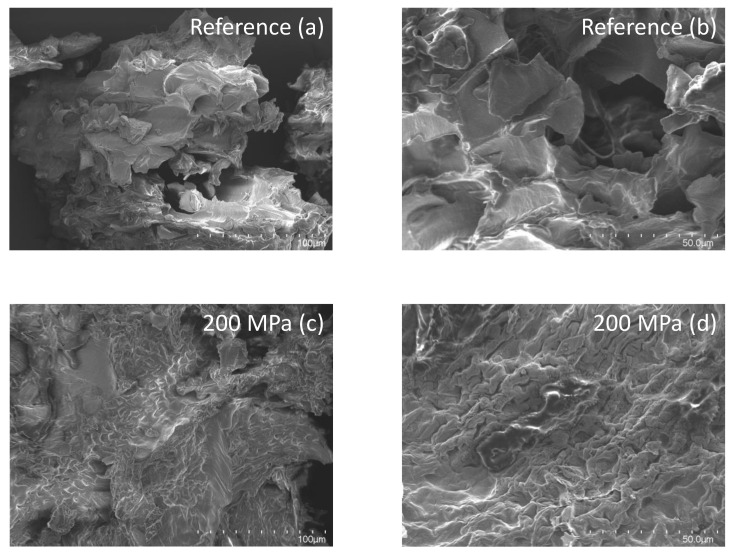
SEM micrographs of okara. (**a**,**b**) Reference, and (**c**,**d**) HPP-treated oat okara at 200 MPa, (**e**,**f**) 400 MPa, and (**g**,**h**) 600 MPa. The magnification of micrographs in the first column was 500× (**a**,**c**,**e**,**g**), whereas the second column had a magnification of 1000× (**b**,**d**,**f**,**h**).

**Table 1 foods-12-03736-t001:** Protocol from the HPP treatments.

Pressure (MPa)	Temperature Interval (°C)	Pressure Interval (MPa)
200	15.3–17.8	200.6–203.9
400	18.3–20.4	399.8–405.2
600	21.4–22.5	600.4–605.0

**Table 2 foods-12-03736-t002:** Settings for the DSC.

	Start (°C)	Limit (°C)	Rate (°C/min)	Hold (min)	Sampling (s)
	25	25	10	1	0.2
End step	25	200	10	0	0.2

**Table 3 foods-12-03736-t003:** Test profile for the RVA.

Time		
00:00	Temp	50 °C
00:00	Speed	960 rpm
00:10	Speed	160 rpm
01:00	Temp	50 °C
06:50	Temp	140 °C
09:20	Temp	140 °C
15:10	Temp	50 °C
17:10	End	

**Table 4 foods-12-03736-t004:** Water content and water activity in reference and all HPP-treated soy okara samples (*n* = 3). No significant differences were found (*p* > 0.05).

	Reference	200 MPa	400 MPa	600 MPa
Water content (%)	77.10 ± 0.06	76.79 ± 0.17	76.76 ± 0.31	76.73 ± 0.06
Water activity	0.993 ± 0.004	0.992 ± 0.001	0.992 ± 0.002	0.991 ± 0.003

**Table 5 foods-12-03736-t005:** Total dietary fiber content including insoluble and soluble fiber content in reference and all HPP-treated soy okara samples on dry basis.

	Reference	200 MPa	400 MPa	600 MPa
Total dietary fiber (%)	26.9 ± 4.0	24.5 ± 3.7	24.4 ± 3.7	24.9 ± 3.7
Insoluble fiber (%)	29.4 ± 4.4	25.1 ± 3.8	24.5 ± 3.7	23.8 ± 3.6
Soluble fiber (%)	n.d.	n.d.	n.d.	1.1 ± 0.2

n.d. = not detectable, *n* = 1, the standard deviation was based on the measuring uncertainty of ±15%.

## Data Availability

The datasets generated during the current study are publicly available from the corresponding author upon request.

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
