# Peer review of "High-Pressure Pasteurization of Soy Okara"

_foods, 2023, doi:10.3390/foods12203736_

Round 1

Reviewer 1 Report

The manuscript entitled “High-Pressure Pasteurization of Soy Okara” is well-written and has scientific merit. However, it needs some improvements: 

Lines 197 & 199: Please be consistent with “min” or “minutes”.

Equation 1-2: Please mention the elaborated term of “??”

Why didn’t you measure all proximate compositions, especially the protein content of HPP-treated Okara, since it is the most important nutrient of Okara?

In the discussion, the authors mainly explained their findings. However, it lacks reasonings (such as mechanism of action) with appropriate references. 

Author Response

Dear Reviewer 1,

Thank you for the valuable comments. We have done our best to make the suggested changes and hope you find them satisfactory in the revised manuscript. Please find our responses to the comments below. In the revised manuscript new and significantly modified sections are indicated with red font.

Comment #1 Lines 197 & 199: Please be consistent with “min” or “minutes”.

Response: Thank you for this observation. We have changed “minutes” to “min” so that it is consistent throughout the manuscript (lines 142 and 217).

Comment #2:  Equation 1-2: Please mention the elaborated term of “??”.

Response: Thank you for this comment, we have now changed from “db” to “dry solid”. Hope this clarifies Equation 1-2 (p. 6, line 221).

Comment #3: Why didn’t you measure all proximate compositions, especially the protein content of HPP-treated Okara, since it is the most important nutrient of Okara?

Response: We focused on the proximate composition of the non-treated soy okara (Table 4) as HPP does not affect the proximate composition as such due to that it is a hermetically closed system where there is no transfer of molecules between the outside and the inside. Proteins can denaturate by HPP treatment (that is why we performed the DSC analysis, Section 3.3), but the content of macromolecules will stay the same. This is why we have not measured the proximate composition again after the HPP treatments.

We measured the content of soluble and insoluble fiber after the HPP treatments, due to that HPP can change the conformation of fibers and thereof change the ratio between soluble and insoluble fiber (p. 10, Table 6). We hope this response satisfies your question.

Comment #4: In the discussion, the authors mainly explained their findings. However, it lacks reasonings (such as mechanism of action) with appropriate references. 

Response: Thank you for this comment. We have added more information about the mechanisms of microbial inactivation in the results. We hope this explains our findings better (p. 7, lines 257-272).

Reviewer 2 Report

The manuscript, entitled “High-Pressure Pasteurization of Soy Okara” described that to investigate the effect of high-pressure pasteurization at 200, 400, and 600 MPa on the shelf life of soy okara. Functional properties were also evaluated to explore how the thermal properties, viscosity, and water and oil holding capacities were affected after the HPP treatment at different pressures. Due to some flaws, my suggestion is major revision.

Comments:

1. In the first section of results and discussion, the growth of total aerobic count, yeast and mold, and lactic acid bacteria was detected. And the authors stated that bacterial spores are difficult to inactivate with HPP, and the treatment can trigger spore germination. Did the authors consider about VBNC, which can also be inactivated in the production process of soy beverage, and it can be reactive in the okara.

2. What are the main genera or species of bacteria that cause okara spoilage? Did the author detect about it?

3. Besides the HPP method, what other methods can be used to process okara? Why did the author choose HPP, and what are the advantages over other methods? It is suggested to discuss about it in the section of results and discussion.

4. In the table 5, n=3, but in table 6, the authors did not mention the number of Parallel experiment number.

5. There is a mistake in the section of 3.1. Microbiological content, the authors clarified that the growth of lactic acid bacteria was low in both the reference and the crude soy okara (Figure 2c) and no significant difference could be detected. But the following P value is less than 0.05, and I think it should be greater than 0.05.

6. Please keep the format of references consistent.

Author Response

Dear Reviewer 2,

Thank you for the valuable comments. We have done our best to make the suggested changes and hope you find them satisfactory in the revised manuscript. Please find our responses to the comments below. In the revised manuscript new and significantly modified sections are indicated with red font.

Comment #1: In the first section of results and discussion, the growth of total aerobic count, yeast and mold, and lactic acid bacteria was detected. And the authors stated that bacterial spores are difficult to inactivate with HPP, and the treatment can trigger spore germination. Did the authors consider about viable but non-culturable (VBNC), which can also be inactivated in the production process of soy beverage, and it can be reactive in the okara.

Response: Thank you for this comment, we had not considered VBNC, but this is a very interesting food safety aspect of HPP. We have added this consideration in the discussion in the results (p. 7, lines 298-299) and in the conclusions (p. 16, lines 451-453) in the manuscript.

Comment #2: What are the main genera or species of bacteria that cause okara spoilage? Did the author detect about it?

Response: This was our first study investigating general microbial content in soy okara. Other members of the research group have an ongoing study in our lab based on these results where genera and species are in focus. The only specific species of bacteria that we discussed in this study was Bacillus Cereus as it is a common spore-forming bacteria among legumes. We do not dare to make any further hypotheses about what more specific bacteria that could be found in the soy okara as we did not make any further analysis on this. We have written about the type of bacteria and microorganisms that has been found in other studies in the introduction (Enterobacteriaceae, Bacillus cereus, yeasts, molds, and mesophilic bacteria), (p. 2, lines 58-59).

Comment #3: Besides the HPP method, what other methods can be used to process okara? Why did the author choose HPP, and what are the advantages over other methods? It is suggested to discuss about it in the section of results and discussion.

Response: We have found patents on other processes to treat soy okara which we have mentioned in the introduction (p. 2, lines 79-84). We are also writing about the advantages of HPP when it comes to the preservation of nutrients and quality of taste, color, odor, and texture (p. 2, lines 85-87). Due to the high viscosity of soy okara, HPP can also be more beneficial as it does not require any pumping or stirring which is an advantage over other pasteurization techniques, such as plate-heat exchanger and scraped surface heat exchanger. We have added this information in the introduction (p. 2, lines 91-93).

To the author’s knowledge, there is no other study that has looked at other pasteurization methods, creating a shelf-life study, similar to our study. Some studies have been found where soy okara has been heat-pasteurized before fermentation, but those results are difficult to compare to ours.

Comment #4: In the table 5, n=3, but in table 6, the authors did not mention the number of Parallel experiment number.

Response: Thank you for your comment. Only one replicate was performed on this analysis as it was sent for external analysis at an accredited lab (Eurofins). This is mentioned in the method section (p. 5, lines 176-177) The standard deviation is based on the measuring uncertainty of ± 15 %.

We have added more information about the number of replicates and what the standard deviation is in Section 3.2, Table 6 (p. 10, line 338).

Comment #5: There is a mistake in the section of 3.1. Microbiological content, the authors clarified that the growth of lactic acid bacteria was low in both the reference and the crude soy okara (Figure 2c) and no significant difference could be detected. But the following P value is less than 0.05, and I think it should be greater than 0.05.

Response: Thank you for this observation. We have changed to P > 0.05 (p. 6, line 246).

Comment #6: Please keep the format of references consistent.

Response: Thank you, the format of the references in the text in the manuscript has been corrected. In the reference list, we have added information about the access date to all links, and have double-checked the format.

Reviewer 3 Report

Dear Author, I reviewed the manuscript (foods-2607074) entitled High-Pressure Pasteurization of Soy Okara. This manuscript presents relevant information about the pasteurization process applied to soy okara. However, some sections of the presented data can be improved. For this reason, I consider that this manuscript needs minor changes to be considered for publication in this journal. 

Additional comments.

Highlight the advantages of using high-pressure pasteurization to elaborate functional foods using soy okara.

Check the paragraphs extension in this manuscript.

Include an experimental design that contains statistical factors and variables of response in the statistical analyses applied to the findings of this research.

Compare these findings with similar assays where high-pressure pasteurization was applied to soy byproducts. 

Include future trends to keep working with the obtained data. 

Try to conclude with a general statement of the most relevant part of this study.

Author Response

Dear Reviewer 3,
Thank you for the valuable comments. We have done our best to make the suggested changes and hope you find them satisfactory in the revised manuscript. Please find our responses to the comments below. In the revised manuscript new and significantly modified sections are indicated with red font.

Comment #1: Highlight the advantages of using high-pressure pasteurization to elaborate functional foods using soy okara.

Response: Thank you for this comment, in the manuscript we have mentioned other studies where HPP has been used to increase the percentage of soluble fiber or to improve the extraction yield of proteins and fibers which in turn could be used for various functional foods (p. 3, lines 111-114). In the viscosity results, we write about the increase of soluble fiber that seems to increase the viscosity which in turn increases the ability to form gels (p. 11, lines 377-379).

We have added information in the introduction about how HPP can affect proteins and in turn change various functional properties in foods (p. 3, lines 109-111).

Comment #2: Check the paragraphs extension in this manuscript.

Response: Thank you for this observation, we have now corrected line and paragraph spacing and indents.

Comment #3: Include an experimental design that contains statistical factors and variables of response in the statistical analyses applied to the findings of this research.

Response: In the microbiological analysis, water content, water activity, fiber analysis, WHC, and OHC we have performed Tukey test and Games-Howell test to evaluate significant differences (described in section 2.10, p. 6, line 231-234).

We have added more information about how we did the statistical comparisons in the microbiological analysis in figure legend 2, as we understand that this was not very clear (p. 9, lines 306-308).

Comment #4: Compare these findings with similar assays where high-pressure pasteurization was applied to soy byproducts. 

Response: Thank you for your comment. When searching in the literature, high pressure has mostly been applied to soy byproducts to increase protein extraction yield, or to improve the functionality of the dietary fiber which we have written about in the introduction (p. 3, lines 111-113). We have not found any study where high pressure has been applied to soy okara with the objective of extending its shelf life and is therefore difficult to be able to compare with other studies.

However, we have now added a section where we compare our microbial results with another study applying high hydrostatic pressure on soaked oats and rice in section 3.1 (p. 7, lines 273-282), which we find are relatively similar products to soy okara. We hope this adds value and gives a better picture.

Comment #5: Include future trends to keep working with the obtained data. 

Response: Thank you, we have added a future trends section in the conclusions which we hope is satisfactory (p. 15-16, lines 446-453).

Comment #6: Try to conclude with a general statement of the most relevant part of this study.

Response: After the future work paragraph we continued with general statements of the most relevant parts of our study (p. 16, lines 453-459).

Reviewer 4 Report

·         Line 46, page 1: “which gives rise to approximately 10 million tons of soy okara every year on the global scale”, add reference o for this.

·         Line 50, page 2: “marzipan”, write the meaning of this in bracket.

·         Line 57, page 2: “while Salmonella and Listeria monocytogenes are not present”…delete this portion.

·         Line 80-82, page 2: However, there is an alternative pasteurization method that can be beneficial for the microbial stabilization and quality of soy okara, namely high-pressure pasteurization (HPP)”, in previous line you described the thermal processing a costlier process then how you are suggesting HPP procedure?.. Is that cheaper for small scale soya processors?

·         Line 124, page 3: “…for at least 24 hours and thereafter thawed in a water bath”, please mention the thawing temperature.

·         Line 129, page 3: “The samples were pasteurized for 3 minutes at three different pressures, 200, 400, and 600 MPa:, please mention the reference from which the method of soy okara pasteurization has been taken.

·         Line 135, page 3: “will be referred to as references”, please refer this as control instead of reference.

·         Line 214, page 6: “Tukey test was performed, and for non-parametric data sets, a Games-Howell test”, please add references for both the test.

·         Line 241, page 6: “A safe food product should have limited growth of total aerobic count, with a maximum microbial load of log 5-6 cfu/g”, please try adding soy related reference instead any gross food reference.

·         Table 5: If there is no significant difference in between the values, there is no need to add superscript in each value, remove superscript “a” (* Data with different letters are significantly different, p < 0.05, n = 3????).

·         Table 6: Mention the probable reasons of not detection of soluble dietary fiber in control, 200MPa and 400MPa?

·         Reference 12: is that complete? Check once.

·         Reference 28 & 30: Refer the link to that online available data.

Author Response

Dear Reviewer 4,

Thank you for the valuable comments. We have done our best to make the suggested changes and hope you find them satisfactory in the revised manuscript. Please find our responses to the comments below. In the revised manuscript new and significantly modified sections are indicated with red font.

Comment #1: Line 46, page 1: “which gives rise to approximately 10 million tons of soy okara every year on the global scale”, add reference o for this.

Response: Thank you for your comment, we have clarified where the 10 million tons of soy okara comes from in the manuscript (p. 2, lines 47-48). It was based on our calculation of 9.2 million tons of soybeans processed into tofu and soy beverages times 1.1 kg soy okara produced for every processed kilogram soybean (9.2 million tons x 1.1 kg soy okara ≈ 10 million tons okara).

Comment #2:  Line 50, page 2: “marzipan”, write the meaning of this in bracket.

Response: The confectionary ingredient “marzipan” has been clarified in brackets (p. 2, lines 51-52).

Comment #3: Line 57, page 2: “while Salmonella and Listeria monocytogenes are not present”…delete this portion.

Response: Thank you for your comment, the line has been removed (p. 2, lines 59-60).

Comment #4: Line 80-82, page 2: However, there is an alternative pasteurization method that can be beneficial for the microbial stabilization and quality of soy okara, namely high-pressure pasteurization (HPP)”, in previous line you described the thermal processing a costlier process then how you are suggesting HPP procedure?.. Is that cheaper for small scale soya processors?

Response: In the manuscript, we have one example where the energy use is lower for HPP compared to a retort (p. 2, lines 93-96), but the costs are of course very much dependent on the type of product, packaging, and processing. HPP is also known for its high investment and maintenance costs which can make it difficult to implement by smaller scale producers. We have tried to clarify this in the manuscript and hope it gives a better overview of the costs (p. 2-3, lines 96-100).

Comment #5:  Line 124, page 3: “…for at least 24 hours and thereafter thawed in a water bath”, please mention the thawing temperature.

Response: Thank you for this comment, we applied water with a starting temperature of approximately 50 °C and let it stand to thaw. We have added this information to the manuscript (p. 3, line 137).

Comment #6: Line 129, page 3: “The samples were pasteurized for 3 minutes at three different pressures, 200, 400, and 600 MPa:, please mention the reference from which the method of soy okara pasteurization has been taken.

Response: We chose the HPP processing time and pressure based on the company HPP Nordic's experience for a wide range of foods. A processing time of 3 minutes was chosen as it is a typical commercial holding time for HPP in the industry. Pressures between 300-600 MPa are also common in the industry. To make the study industrially relevant we chose to use similar parameters. Therefore, we have no reference to the HPP protocol used. We have added a line in the manuscript mentioning that 3 minutes is a typical commercial holding time to make it easier for the reader to understand why we chose 3 minutes (p. 3, lines 142-143).

Comment #7: Line 135, page 3: “will be referred to as references”, please refer this as control instead of reference.

Response: Thank you for this suggestion. In the present study, we have both analytical results regarding physicochemical and functional analysis as well as microbiological results. For the analytical results, we believe that “reference” is a more correct term, but we agree that “control” is a better word for the microbiological analysis. However, for the sake of consistency, we choose to use “reference” throughout the whole manuscript to not confuse the reader since it is the same material we are referring to (non-treated okara).

Comment #8: Line 214, page 6: “Tukey test was performed, and for non-parametric data sets, a Games-Howell test”, please add references for both the test.

Response: We have now added references to the the statistical tests that we performed (p. 6, lines 233 and 234).

Comment #9: Line 241, page 6: “A safe food product should have limited growth of total aerobic count, with a maximum microbial load of log 5-6 cfu/g”, please try adding soy related reference instead any gross food reference.

Response: Thank you for your comment, due to that soy okara is not a commercial product yet we found it difficult to find any relevant soy-related reference. For example, soy milk and soy protein isolates are very different products compared to soy okara when it comes to moisture content and viscosity. The limits we have found already are mostly used as an indication of the amount of bacteria that would be reasonable for a commercial product or not. However, this must of course be further investigated, and it will be important to know more specifically the type of bacteria that can grow in the soy okara to be able to evaluate the microbial safety better.

Comment #10: Table 5: If there is no significant difference in between the values, there is no need to add superscript in each value, remove superscript “a” (* Data with different letters are significantly different, p < 0.05, n = 3????).

Response: We have now removed the superscript “a” and has instead added the information about no significant difference between the samples in table legend 5 (p. 9, lines 322-323).

Comment #11: Table 6: Mention the probable reasons of not detection of soluble dietary fiber in control, 200MPa and 400MPa?

Response: Thank you, we have added two possible reasons for this in the manuscript (p. 10, lines 332-334).

Comment #12:  Reference 12: is that complete? Check once.

Response: We checked reference 12, and it should now be complete with page numbers.

Comment #13:  Reference 28 & 30: Refer the link to that online available data.

Response: Reference 28, which is a patent, has now a proper link to the available data. Reference 30 was found in the book “High-pressure processing of food: Principles, technology and applications”. We have now added this book reference as well on p. 2, line 96 where reference 30 has been used to clarify where the information comes from. We saw that reference 30 was difficult to find online.

Round 2

Reviewer 2 Report

1. In the section of the Conclusion, the authors clarified that “. Due to the denaturation of most proteins in the soy okara during the production process of soy beverage, the HPP treatment did not have any additional impact on the protein structure”, however, in the study the authors did not detect if the protein structure changed. Please provide the references.

2. It is suggested to add the discussion compared HPP with other Sterilization methods. And compared with other Sterilization methods, what’s the advantages of HPP?

3. What is the size of the treatment at one time for the HPP method? How much weight of Soy Okara can be processed at a time?

4. In line 137, the blank space between “50” and “℃” should be deleted.

English writing level can be improved.

Author Response

Reviewer number 2
Thank you for the valuable comments. We have done our best to make the suggested changes and hope you find them satisfactory in the revised manuscript. Please find our responses to the comments below. In the revised manuscript new and significantly modified sections are indicated with blue font.

Comments and Suggestions for Authors

Comment #1: In the section of the Conclusion, the authors clarified that “. Due to the denaturation of most proteins in the soy okara during the production process of soy beverage, the HPP treatment did not have any additional impact on the protein structure”, however, in the study the authors did not detect if the protein structure changed. Please provide the references.

Response: Thank you, we have now rephrased and corrected that sentence (p. 15, lines 441-443). What we wanted to say was that most soy proteins were denatured already in the soy beverage production, due to elevated temperatures (up to 100 °C). Therefore, there are not many soy proteins left for HPP to denature. However, the protein structure might of course have been altered by HPP but we have not investigated that in detail.

We also added a sentence about the temperature that is reached in the soy beverage production in the results of thermal properties to increase clarity (p. 10, lines 361-362).

Comment #2: It is suggested to add the discussion compared HPP with other Sterilization methods. And compared with other Sterilization methods, what’s the advantages of HPP?

Response: Thank you for this valuable comment, we have added a comparison with a similar study investigating microbial load in soy okara in a shelf-life study design using thermal treatment at 80 and 200 °C. HPP had a larger reduction of microbial activity for yeast and mold, but the incubation temperature differed between our study and the cited study which has been commented in the manuscript (p 7, lines 258-267). We have also added some more information in the discussion about the advantage of combining HPP and thermal treatment (p. 7, lines 302-307).

Comment #3: What is the size of the treatment at one time for the HPP method? How much weight of Soy Okara can be processed at a time?

Response: Thank you for our comment, information regarding the batch size (200 g) in our study can be found on p. 3, line 141. We have added information about available batch sizes in the industry in the introduction and a possible weight of soy okara that could be processed in one batch (p. 2, lines 91-94).

Comment #4: In line 137, the blank space between “50” and “℃” should be deleted.

Response: Thank you, at line 140 (line 137 in the former version of the manuscript), the marginal adjustment gives rise to a perceived double spacing, but there is only one space between “50” and “℃”.

Comments on the Quality of English Language

English writing level can be improved.

Response: Thanks for the comment. The manuscript will be checked for grammar mistakes by a native speaker.